# Potential present and future distributions of the genus *Atta* of Mexico

**Jorge A. Gómez-Díaz**[1,2]*, **Martha L. Baena**[1], **Arturo González-Zamora**[1], **Christian A. Delfín-Alfonso**[1,3]*

**1** Instituto de Investigaciones Biológicas, Universidad Veracruzana, Xalapa, Veracruz, Mexico, **2** Centro de Investigaciones Tropicales, Universidad Veracruzana, Xalapa, Veracruz, Mexico, **3** Laboratorio de Zoología, Instituto de Investigaciones Biológicas, Universidad Veracruzana, Xalapa, Veracruz, Mexico

* jorggomez@uv.mx (JAGD); cdelfin@uv.mx (CADA)

**Data Availability Statement:** All relevant data are within the paper and its Supporting Information files.

**Funding:** The author(s) received no specific funding for this work.

## Abstract

Temperature and precipitation influence insect distribution locally and drive large-scale bio-geographical patterns. We used current and future climate data from the CHELSA database to create ensemble species distribution models for three *Atta* leaf-cutting ant species (*Atta cephalotes*, *A. mexicana*, and *A. texana*) found in Mexico. These models were used to estimate the potential impact of climate change on the distribution of these species in the future. Our results show that bioclimatic variables influence the distribution of each *Atta* species occupying a unique climatic niche: *A. cephalotes* is affected by temperature seasonality, *A. mexicana* by isothermality, and *A. texana* by the minimum temperature of the coldest month. *Atta texana* and *A. mexicana* are expected to decline their range by 80% and 60%, respectively, due to rising temperatures, decreased rainfall, and increased drought. Due to rising temperatures and increased humidity, *Atta cephalotes* is expected to expand its range by 30%. Since *Atta* species are important pests, our coexistence with them requires knowledge of their ecological functions and potential future distribution changes. In addition, these insects serve as bioindicators of habitat quality, and they can contribute to the local economy in rural areas since they are eaten as food for the nutritional value of the queens. In this sense, presenting a future perspective of these species' distribution is important for forest and crop management. Education programs also are necessary to raise awareness of the importance of these ants and the challenges they face because of climate change. Our results offer a perspective of climate change studies to define conservation and adaptation strategies for protecting vulnerable areas such as high-elevation remnant forests.

## Introduction

Ants are the most successful insect group in terrestrial environments, with a diversity of almost 13,000 species [1]. The group of Mexican ants, though not extensively studied [2], encompasses various species of leaf-cutting ants belonging to the *Atta* genus (Formicidae), known as prominent herbivores in the Neotropical region [3]. Nonetheless, substantial progress has been made in addressing these knowledge deficiencies, exemplified by the research conducted by Dáttilo et al. [4]. Their study diligently gathered all existing data on ants across Mexico.

**Competing interests:** The authors have declared that no competing interests exist.

They discerned key patterns within the compiled dataset and identified geographical areas where further sampling endeavors should be focused.

In Mexico, three species of *Atta* are recognized in two groups: *A. mexicana* and *A. texana* (both in *Archeatta*) and *A. cephalotes* (*Atta sensu stricto*). Biogeographic analyses suggest that Mexico and South America form the distribution range of their most recent common ancestor [5]. These three species live in the open canopy of tropical and subtropical environments. *Atta* ants, often called leafcutter ants, play essential ecological roles in ecosystems, particularly in nutrient cycling and soil aeration [6]. Their intricate underground colonies contribute to soil turnover [6]. However, they are also renowned as significant agricultural pests, with their voracious foraging behavior leading to the defoliation of a wide range of plant species, causing notable economic impacts in affected regions [7]. *Atta* species utilize all the leaves they cut to cultivate fungi in their underground chambers [6].

The relationship between *Atta* ants and fungi is a well-documented symbiotic interaction. *Atta* ants carry fresh leaves into their nests and chew them up as a substrate to cultivate the *Leucoagaricus gongylophorus* fungus, which they feed [3, 8]. The fungus breaks down the leaves into nutrients the ants can absorb [9]. This relationship is mutually beneficial, as the ants provide the fungus with a food source, and the fungus helps the ants to digest their food [3]. Higher air temperatures are more suitable for the brood and fungal chambers, leading to more frequent nest establishment [10].

Species distribution modeling has been used to identify the environmental conditions necessary for a species to survive and thrive over a long period [11, 12]. But this methodology has rarely been used for ants since only a few studies have addressed the conservation of these species [13]. The data collected from this study of three *Atta* species' spatial distribution can elucidate how they developed in conditions that differed from the present. Given *Atta* ants' unique behavior of cultivating fungi using cut leaves, their challenges to agriculture cannot be understated. Recognizing their ecological significance and impact on agriculture, this study seeks insights crucial for effective colony control and strategic management, as reviewed by [6].

In general, species of mountainous areas, such as the Mexican *Atta*, are at high risk of being harmed by climate change [14]. Climate change is causing various species to move to new areas for suitable climatic niches [12, 14]. Even though Mexican *Atta* species are relatively resilient to deforestation and habitat fragmentation, climate change is a major threat to their survival [12]. Climate change is expected to cause the distribution of Mexican *Atta* species to shift, fragment, and become increasingly unsuitable for these ants [12, 14]. Therefore, this study aimed to determine how climate change will affect the distribution of Mexican *Atta* species in the future. Particularly, we studied: i) the current distribution of each species, ii) the factors in the environment that have the highest contribution to the ecological niche of each species, and iii) the future climate change effects on each species.

## Materials and methods

### Species distribution models

We collected data on the distribution of *Atta* species in the field to create spatial distribution maps. Also, we used the Mexican ant database [4] and updated it with data from the Global Biodiversity Information Facility [15–17]. We found 7326 occurrences (3640 for *A. cephalotes*, 1575 for *A. mexicana*, and 2111 for *A. texana*). The coordinates were checked for accuracy and completeness using the *clean_coordinates* function from the *CoordinateCleaner* R package [18] to have only one occurrence per km$^2$ (625 records of *A. cephalotes*, 1127 of *A. mexicana*, and 750 of *A. texana*), which were then entered into the distribution model of each species (S1 Dataset).

The climatic data used to create the models was derived from 19 bioclimatic variables from the CHELSA v.1.2 database (https://chelsa-climate.org/; [19]). These variables were used at a resolution of 30 arc seconds (~1 km on the ground). For the present and the future scenarios of climate change for 2086 (average between 2071–2100; hereafter referred to as future), we considered identical bioclimatic factors. To project future climate, we developed four species distribution models for each species, utilizing four distinct global circulation models (GFDL, IPSL, MPI, and MRI) from the Coupled Model Intercomparison Project 6, based on the Shared Socioeconomic Pathway (SSP) 5–8.5 scenario of climate change of the Intergovernmental Panel on Climate Change (IPCC; [12]). We used only the SSP 5–8.5 scenario (high GHG emissions: $CO_2$ emissions tripled by 2075) since global emissions have increased steadily throughout the 21st century [12, 20]. Therefore, if a species can survive the proposed scenario, it will likely survive future climate change [12].

The Nature Conservancy's terrestrial ecoregions, based on those created by Olson et al. [21], were used to calibrate a model of the variables (M space; [12, 22]). For each species, ecoregions were selected where their distribution was known [12] (S1 Table). The environmental variables were also screened for collinearity using a variance inflation factor (VIF) analysis. This analysis is used to identify variables that are highly correlated with each other. Variables with a VIF value greater than 10 were removed from the study using the exclude function from the R *usdm* package vr. 1.1–18 [12, 22]. Values of VIF greater than 10 indicate a high degree of multicollinearity in the model [23]. Of the 19 environmental variables, only seven did not have high collinearity (VIF values < 10) for both *A. cephalotes* and *A. texana* and nine for *A. mexicana* (Table 1; S1 Table).

The *ensemble_modelling* function in the R package *SSDM* was used to create an ensemble species distribution model (ESDM) for each species using a combination of the algorithms of MaxEnt (with the default parameters), support vector machines (SVM), and artificial neural networks (ANN; [12]). Our study employed pseudo-absence data, as is standard in species distribution models (SDMs) when true absence data is unavailable [24]. We used algorithms, specifically MaxEnt, SVM, and ANN, which necessitate pseudo-absences, adhering to recognized

**Table 1. Bioclimatic variables used to predict the distribution of three Mexican *Atta* leaf-cutting ant species and their importance (%).**

| Variable | *A. cephalotes* | | | | *A. mexicana* | | | | *A. texana* | | | |
|---|---|---|---|---|---|---|---|---|---|---|---|---|
| | (n = 625) | | | | (n = 1127) | | | | (n = 750) | | | |
| | % | Min | Max | Mean ± SD | % | Min | Max | Mean ± SD | % | Min | Max | Mean ± SD |
| Mean diurnal range (bio 2) | 21 | 1 | 13 | 6.65 ± 1.7 | 8 | 1 | 16 | 11.48 ± 2.6 | 12 | 2 | 14 | 9.87 ± 1.2 |
| Isothermality (bio 3) | | | | | 18 | 17 | 74 | 57.81 ± 7.3 | | | | |
| Temperature seasonality (bio 4) | 27 | 14 | 228 | 91.43 ± 62.6 | | | | | | | | |
| Maximum temperature of the warmest month (bio 5) | | | | | 9 | 20 | 41 | 30.63 ± 3.4 | | | | |
| Minimum temperature of the coldest month (bio 6) | | | | | | | | | 22 | 3 | 18 | 7.16 ± 2.3 |
| The mean temperature of the wettest quarter (bio 8) | 26 | 3 | 28 | 23.43 ± 2.9 | 11 | 13 | 34 | 23.03 ± 3.7 | 15 | 10 | 29 | 24.45 ± 4.4 |
| The mean temperature of the driest quarter (bio 9) | | | | | | | | | 15 | 7 | 30 | 21.29 ± 7.5 |
| Precipitation of wettest month (bio 13) | 14 | 115 | 1130 | 398.9 ± 161.5 | 14 | 30 | 799 | 209.9 ± 94.9 | 9 | 57 | 542 | 121.89 ± 29.4 |
| Precipitation of driest month (bio 14) | 2 | 1 | 541 | 82.20 ± 70.9 | 7 | 0 | 93 | 9.948 ± 13.5 | | | | |
| Precipitation seasonality (bio 15) | 3 | 8 | 117 | 48.89 ± 17.4 | 11 | 19 | 126 | 95.11 ± 17.7 | 16 | 11 | 108 | 27.94 ± 12.4 |
| Precipitation of warmest quarter (bio 18) | 8 | 55 | 2322 | 596.6 ± 344.6 | 12 | 4 | 1177 | 263.6 ± 219.0 | 12 | 102 | 816 | 208.07 ± 67.0 |
| Precipitation of coldest quarter (bio 19) | | | | | 11 | 5 | 723 | 67.19 ± 63.2 | | | | |

Relative niche values (based on geographic records) in the ecological dimension (mean values, standard deviations, maximum and minimum values) for each ensemble species distribution model are presented.

ecological modeling practices. The sensitivity-specificity equality (SES) metric was used to generate binary maps of each species, as recommended by Liu et al. [25]. In the case of the four future models per species, the binary maps were summed to consider the variation in the values of the predictors and have a measure of uncertainty in these projections. The performance of the models was evaluated using the area under the curve (AUC) and the Partial-ROC test at 5% [26]. The AUC measures the overall predictive accuracy of a model, while the Partial-ROC test at 5% measures the model's accuracy at a specific threshold. The referenced 5% pertains to the allowable omission error in the models assessed [26]. Specifically, models that omit known presence points more than 5% of the time are deemed less reliable. This threshold was chosen based on the error characteristics of the occurrence data used. The statistical analyses in this study were conducted using the R v.3.6.3 programming language [27].

## Climatic niches

We calculated each species' climatic niche breadth, extracting climatic data from the present shared variables among the three species to investigate how the range of climatic conditions a species can tolerate relates to the environmental and geographic factors influencing its distribution [28]. Then we performed a principal components analysis (PCA) with the function *prcomp* and obtained the first three axes to create the ecological background. Next, we plotted the coordinates of the three axes in a tridimensional space using the functions *plot3d* and *ellipse3d* of the R package *rgl* vr. 0.111.6 [29], assuming that species' fundamental ecological niches are convex in shape [28, 30].

## Results

The distribution models of *A. cephalotes*, *A. mexicana*, and *A. texana* were highly supported. The AUC and partial ROC values for each model were all statistically significant, indicating that the models were accurate in their predictions. The mean AUC values for the three models were 0.903, 0.902, and 0.938, respectively (Table 2). The mean partial ROC values for the three models were 1.355, 1.439, and 1.670, respectively. These results suggest that the distribution models of all three species are accurate and can be used to predict their future distributions. Although the studied species live nearby (Fig 1), each occupies a distinct ecoregion and climatic niche. However, in a small portion of the territory in Mexico, the species *A. mexicana* and *A. cephalotes* share a fraction of climatic variables, suggesting possible spatial sympatry in their distribution. In this scenario, the ESDM for each species varies regarding the significance of the bioclimatic variables that influence their distribution. For example, for *A. cephalotes*, the bioclimatic variables of the highest importance were Temperature Seasonality (Bio 4, 27%), Mean Temperature of Wettest Quarter (Bio 8, 26%), and Mean Diurnal Range (Bio 2, 21%);

**Table 2. Thresholds established to categorize the suitability and evaluation metrics applied to the ESDM for each Mexican *Atta* leaf-cutting ant species studied.**

| Metric | *A. cephalotes* | *A. mexicana* | *A. texana* |
|---|---|---|---|
| Threshold | 0.512 | 0.633 | 0.646 |
| AUC | 0.903 | 0.902 | 0.938 |
| Omission rate | 0.178 | 0.269 | 0.235 |
| Sensitivity | 0.822 | 0.804 | 0.877 |
| Specificity | 0.821 | 0.721 | 0.747 |
| The proportion of correctly predicted occurrences | 0.821 | 0.731 | 0.765 |
| Calibration | 0.587 | 0.670 | 0.737 |

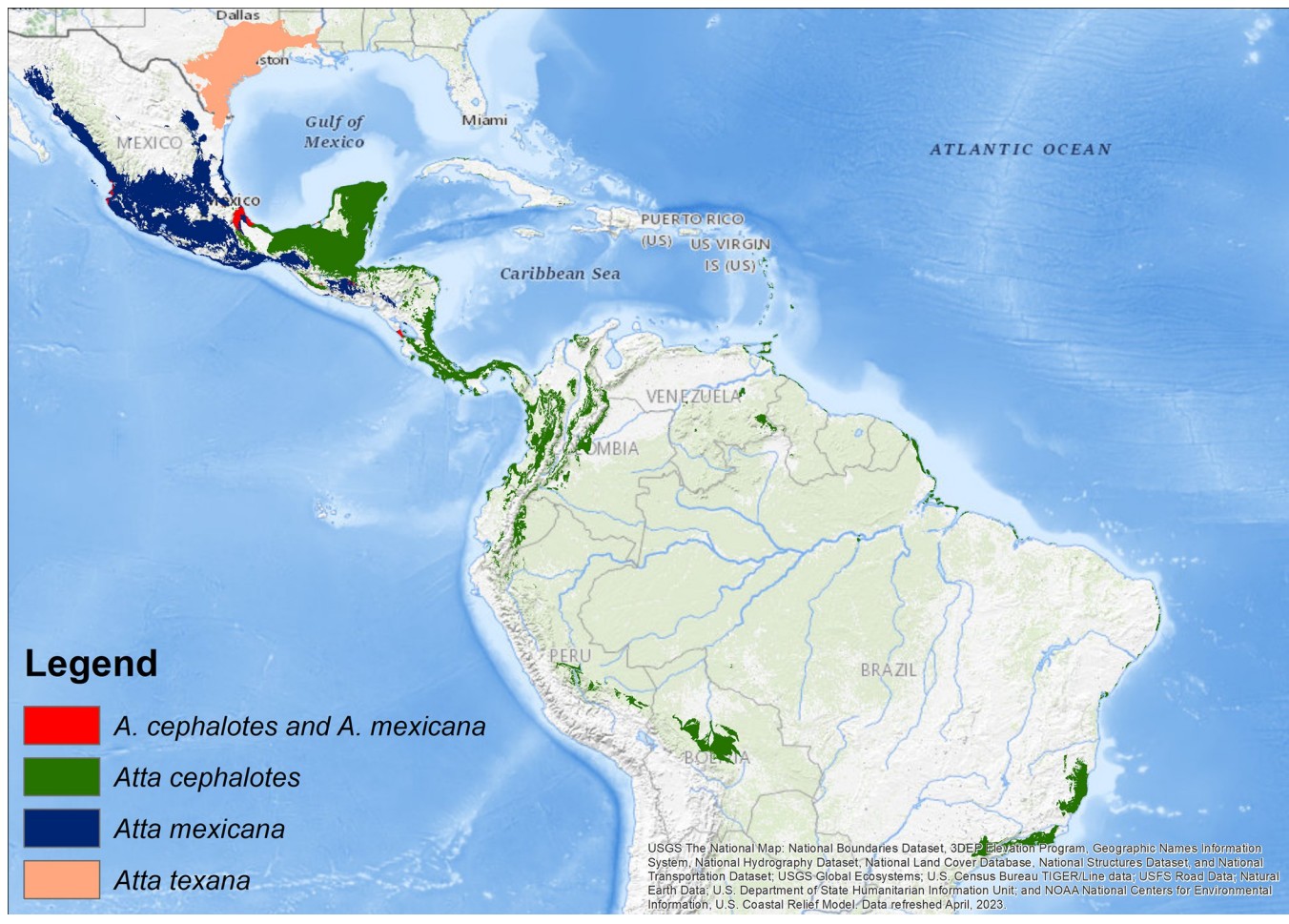

**Fig 1. Distribution models of the studied *Atta* species.** Grey shadow represents topographic relief.

for *A. mexicana*, these were Isothermality (Bio 3, 18%), Precipitation of Wettest Month (Bio 13, 14%) and Precipitation of Warmest Quarter (Bio 18, 12%). For *A. texana*, they were the Minimum Temperature of the Coldest Month (Bio 6, 22%), Precipitation Seasonality (Bio 15, 16%), and Mean Temperature of the Driest Quarter (Bio 9, 15%; Table 1). The variables Mean Diurnal Range, Mean Temperature of Wettest Quarter, Precipitation of Wettest Month, Precipitation Seasonality, and Precipitation of Warmest Quarter (Bio 2, 8, 13, 15, and 18, respectively) were common to all three species.

## Climatic niches

Of the environmental variables selected for each species (*A. cephalotes* = 7, *A. mexicana* = 9, *A. texana* = 7), the most important are the five variables that are shared among them in the modeling process. The ecological niches of the three species of *Atta* leaf-cutting ants varied in size and shape. *A. cephalotes* had the widest climatic niche, while *A. texana* had the narrowest one (Fig 2). Niche overlap was observed among the species, with the *A. cephalotes* ecological niche containing all of the *A. texana* and a portion of the *A. mexicana* niche space. However, the environmental niches of *A. cephalotes* and *A. mexicana* were slightly separated (Fig 2).

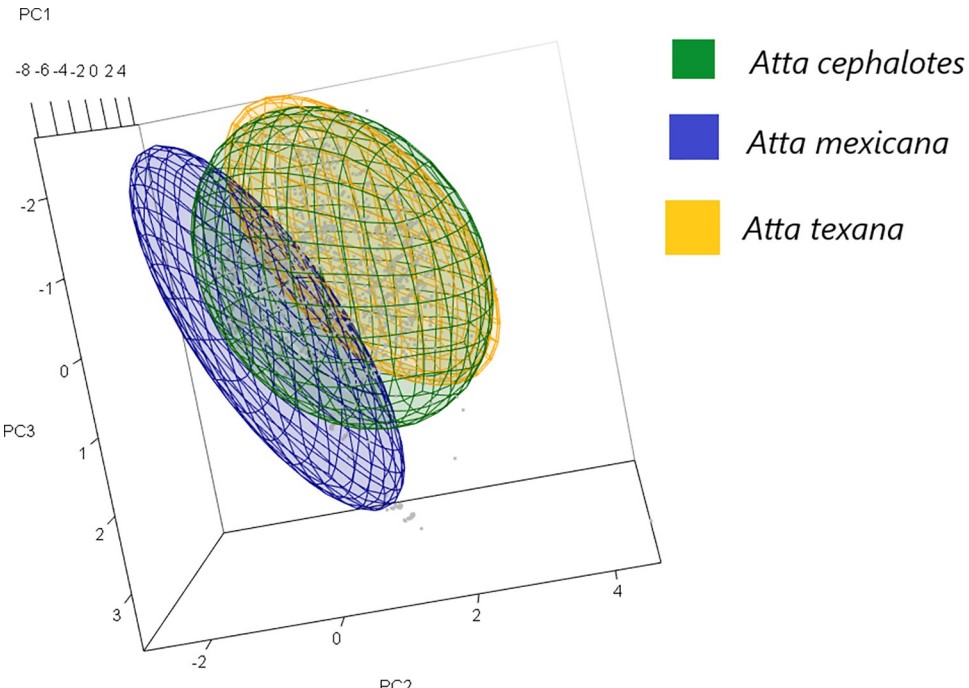

**Fig 2. Exploratory visualization of niche separation.** Niche separation between the *Atta* genus niche's three species (shared variables). A three-dimensional representation was used to visualize the ecological niches of the three *Atta* leaf-cutting ant species based on principal components (PC). Minimum Volume Ellipsoids were used to represent the boundaries of the species' environmental distributions. The green ellipsoid represents *A. cephalotes*, the blue ellipsoid corresponds to *A. mexicana*, and the orange ellipsoid represents *A. texana*. Gray dots represent the environmental conditions where the species have been observed, as determined by the Principal Components Analysis (PCA) of the five shared variables (S2 Table).

A PCA of the ecological data identified three main niche axes that explained 89% of the variation together. The first PC (46%) was positively associated with precipitation of the wettest month (Bio 13), precipitation of the warmest quarter (Bio 18), and negatively with mean diurnal range (Bio 2); this axis can be named "climatic moisture index" and reflects the significance of climatic moisture in this variable and its relevance in the relationship with the mentioned variables. The second PC, which explained 24% of the variation, was associated with precipitation seasonality (Bio 15), while the third PC (19%) was related to the mean temperature of the wettest quarter (Bio 8). The results from niche overlap analysis reveal a significant diversity in the environmental habitats occupied by the three species, which aligns with the findings obtained from the PCA analysis.

*Atta cephalotes* is predominantly found in areas characterized by significantly higher values of Precipitation in the Wettest Month (Bio 13; KW = 820.5, df = 2, p < 0.001), higher values of Precipitation in the Warmest Quarter (Bio 18; KW = 98.5, df = 2, p < 0.001), and intermediate values of Precipitation Seasonality (Bio 15) compared to the other species. *A. mexicana*, on the other hand, inhabit regions with notably higher values of Mean Temperature of Wettest Quarter (Bio 8; KW = 193.03, df = 2, p < 0.001), while exhibiting the lowest values of Precipitation of Wettest Month (Bio 13) and Precipitation Seasonality (Bio 15) when compared to the other species. In the case of *A. texana*, it tends to occupy areas characterized by significantly higher values of Mean Diurnal Range (Bio 2; KW = 190.46, df = 2, p < 0.001), Precipitation Seasonality (Bio 15; KW = 1209.7, df = 2, p < 0.001), and intermediate values of Precipitation of Wettest Month (Bio 13) in comparison to the other species.

## Distribution of the species

The potential current distribution of *A. cephalotes* covered Mesoamerica, Mexico, Colombia, Venezuela, Ecuador, Peru, Bolivia, and Brazil, under the most favorable conditions occurring in the Isthmian-Atlantic Moist Forests and Chocó-Darién Moist Forests in Panama (Fig 3A). Of the three study species, *A. cephalotes* has a southern distribution, including the Atlantic forest in Brazil, with a disjunct distribution. Under a future climate scenario, the geographical range of *A. cephalotes* is expected to increase by 490% (Table 1, Fig 3B). *A. mexicana* current distribution covers central Mexico. The most favorable conditions for the species are found in the Sierra Madre Occidental Pine-Oak Forests and Bajio Dry Forests (Table 1, Fig 3C). However, under the climate change scenario, the models predict a drastic 96% reduction in the species' distribution range (Table 1, Fig 3D). The species practically disappeared except for limited areas on Mexico's Pacific coast (Sonora, Sinaloa, Guerrero, and Oaxaca) and in the central portion of Mexico in the Balsas River Basin. The potential present distribution of *A. texana* covered the area of northern Mexico (Tamaulipas state) and the southeastern USA (Texas and Louisiana; Table 1, Fig 3E). Under the climate change scenario, there is projected to be an 80% reduction in the species' distribution range (Table 1, Fig 3F). The species is eradicated or effectively disappears except for limited areas in Texas and Louisiana.

## Discussion

We found that the three *Atta* species presented spatial separation among them. To the best of our knowledge, this study represents the first investigation focusing on the current and future potential distribution of ants in Mexico so that we will compare our results with similar studies on other insects. For example, different species showing this segregation pattern in Mexico are Melipona stingless bees and the moth in the tribe Arctiini [31, 32]. In ants, this pattern also has been found in the genus *Lasius* in Korea [33] and *Brachyponera nigrita* in Pakistan [34]. This disparity could be attributed to variations in the bioclimatic variables specific to each species [12]. Temperature seasonality was especially important for *A. cephalotes*, Isothermality was for *A. mexicana*, and minimum Temperature of the Coldest Month was for *A. texana*. In the case of *A. texana*, the minimum temperature was also an important variable that explains the future distribution.

We concur that climate change will likely contribute to expanding future distributions of invasive species, including ants [35]. Contrary to general expectations, a study utilizing species distribution modeling for invasive species, considering current and projected climate change conditions (by 2071–2100), revealed that only five species were projected to exhibit an expanded potential distribution (up to 35.8%) with climate change. However, most species were expected to decline, with up to 63.3% reductions. Hence, climate change and invasive ant species must act synergistically as anticipated [35].

Additionally, our findings indicate that future modeling predicts a substantial reduction in the distribution ranges of these species to the extent that they effectively vanish, particularly in the cases of *A. mexicana* and *A. texana*. According to a previous study by Parr and Bishop [36], ant species inhabiting thermally variable microhabitats in tropical regions, such as the canopy and leaf litter environments, are expected to face adverse effects from increasing temperatures. Conversely, species residing in temperate zones and those capable of adjusting their nesting behaviors to avoid elevated temperatures will likely remain unaffected or benefit from the changing climate, as Parr and Bishop [36] reported. This contrasts with the results of the models of these species (*A. mexicana* and *A. texana*) primarily found in mountainous or northern regions characterized by temperate temperatures. These species can counteract the negative effects of climate change by building their nests deeper to escape increasing temperatures.

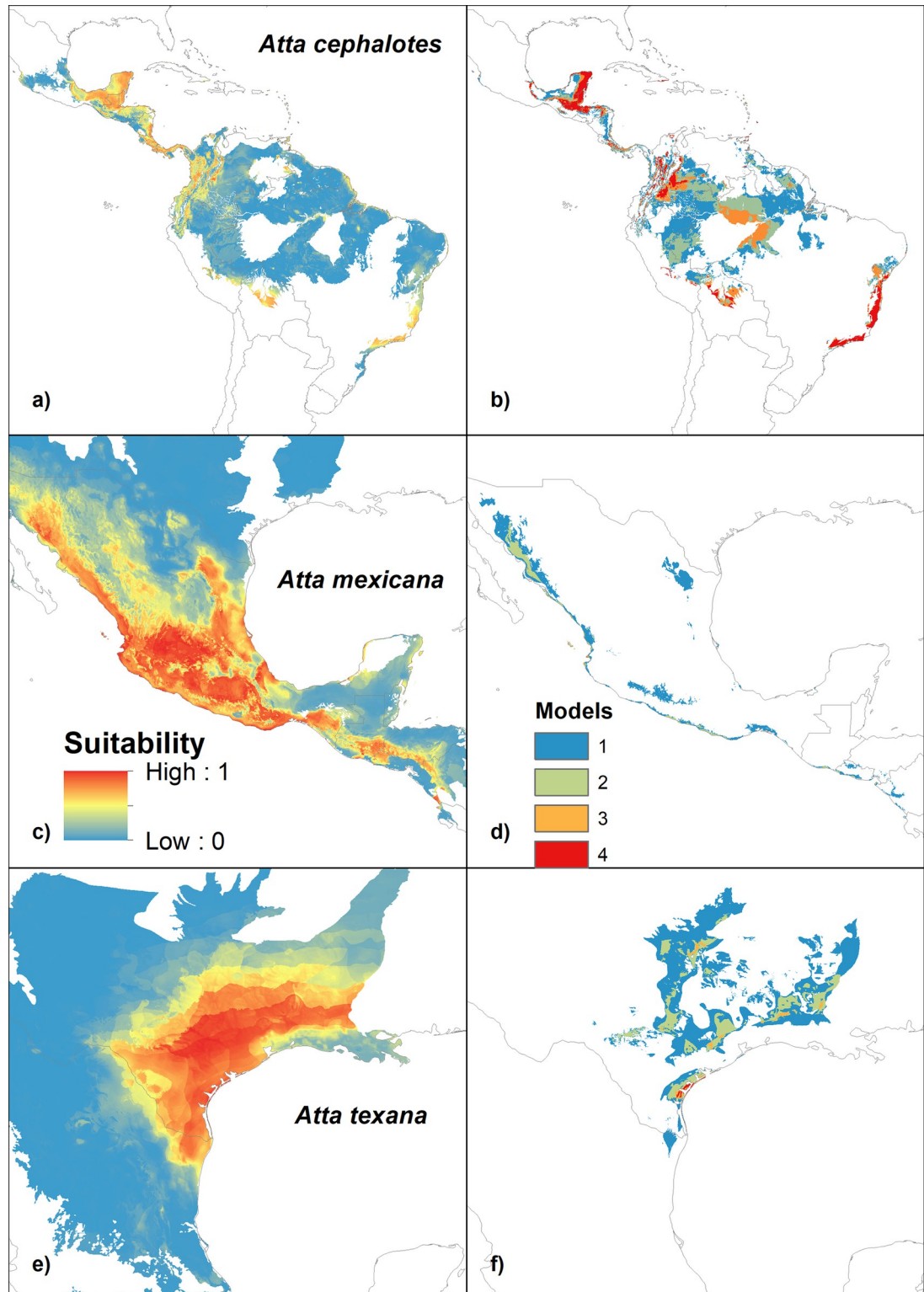

**Fig 3. Potential distribution of *Atta* species.** The possible distribution of *A. cephalotes* in the a) current; b) future; *A. mexicana* in the c) current; d) future; and *A. texana* in the e) current; f) future. In the case of the current scenario, suitability is shown, and in the future scenario, the number of models that predict the distribution is shown.

Climate change models suggest that species loss will be greatest in tropical mountains [12, 37]. Birks [38] estimated that these areas would undergo increased warmth and aridity, with a tendency towards more severe drought conditions in lower mountainous regions. These changes in climate conditions are expected to result in shifts in species distribution, from which some will benefit while others will be harmed [39]. Based on the habitat suitability models, climate change will likely lead to alterations in species distribution, and droughts will probably affect the ant species studied here. Both *A. mexicana* and *A. texana* are typically found in locations characterized by high humidity and temperate temperatures. However, the anticipated change in the geographical distribution of *A. cephalotes* is influenced to a lesser degree, possibly due to the species' existing distribution aligning with warm lowland climates, which could provide an adaptive advantage in the future. Since temperature seasonality seems important for *A. cephalotes*, it is necessary to consider its obligatory interaction with its mutualistic fungus. This is relevant because the temperature is key to the functioning of *Atta* species nests [40]. *Atta* species that inhabit American tropical areas have strict temperature and humidity demands; the fungus cannot survive at $> 30°C$ or $< 10°C$ [41]. Therefore, *A. cephalotes* will expand its distribution toward lowlands that present conditions suitable for building chambers that can maintain fungal gardens in the 20–30°C range. These conditions are typical of low- to mid-elevation rainforests where temperature and humidity exhibit moderate variations throughout the year [42].

In the case of *A. texana*, the model corroborates that the minimum temperature of the coldest month is critical and restricts the spatial distribution of the species. This is the leaf-cutting ant species with the most northerly distribution in the Americas, given the selection for cold-tolerant fungus and desiccation resistance variants. During the winter season, *A. texana* manages the moisture levels of its fungal gardens by actively searching for groundwater and moving the gardens vertically. They relocate the gardens from shallow layers, which have a temperature of approximately 5°C, to deeper layers with temperatures ranging from 10 to 15°C, as observed by Mueller et al. [42]. Another study shows that soil drying in the Sonoran Desert due to climate change has resulted in the demise of numerous *A. texana* nests, restricting the establishment of new nests for 20 years [43]. The estimated 80% reduction in the distribution of *A. texana* in the future indicates that there will be insufficient time to adapt both the fungus's thermal tolerance and the ants' behavior to avoid desiccation.

As temperatures continue to increase according to the current climate change scenario, pests, including *Atta* species, might become more resilient or fortified [44]. Moreover, these ants can be problematic as emitters of biogenic halocarbons [45, 46]. Thus, reducing the distribution of *A. mexicana* and *A. texana* may decrease the emission of these halogases. Nevertheless, the increased distribution of *A. cephalotes* may imply a medium-term risk of emissions of these biogenic halogases, representing an important challenge in reducing emissions of these gases to stop global warming. Future studies should focus on evaluating the economic impacts of *Atta* species and their effects as emitters of greenhouse gases, particularly in vulnerable areas such as high-elevation remnant forests.

Our results indicate that drastic changes in the distribution of *A. texana*, *A. mexicana*, and *A. cephalotes* will be a feature of climate warming. Our findings are important for forest management and conservation because *Atta* species perform key ecological functions depending on the environmental context. They modify the vegetation and plant community structure and are reservoirs of nutrients and organism diversity [6, 47–49]. Furthermore, according to the model used in this study, species such as *A. cephalotes* can spread widely in the Americas and could be used as indicators of forest degradation. Thus, humans could use these ants as part of more efficient farmland management and conservation strategies [5, 50] in Neotropical dry forests, which are considered a conservation priority due to their significant loss of

vegetation cover to agricultural practices, surpassing other terrestrial biomes [5]. Consequently, the dry forests face an elevated risk of pests, such as *Atta* species, becoming more resilient, particularly in response to the increasing temperatures expected to occur under the current climate change scenario [44]. Estimating the ecosystem impact of these ants is a challenge. For this reason, it is important to conduct interdisciplinary studies that address ecological, social, and economic issues that guide us toward their management. Additionally, educational programs are necessary to make it known that *Atta* leaf-cutting ants can be more allies than enemies.

In discussing the outcomes of our study, it is important to acknowledge certain limitations. Firstly, the analysis was conducted based on the available data and variables, which may only encompass part of the spectrum of factors influencing species-environment relationships [51]. This may introduce potential biases and restrict the generalizability of our findings. Additionally, our use of correlative models to infer relationships between climatic variables and the biology/ecology of the target species assumes stationarity. It may only partially capture dynamic relationship shifts over time or under changing environmental conditions [52]. While valuable for making predictions based on observed patterns, these models do not explicitly capture the underlying processes and mechanisms [53]. It is important to exercise caution in applying and interpreting these models, recognizing that correlation does not imply causation and that additional factors may influence the observed relationships.

We acknowledge the limitations of correlative models and encourage further exploration of complementary approaches, such as mechanistic models, to enhance our understanding of species-environment interactions and ecological dynamics [54]. Moreover, it is crucial to recognize that our study focused on a specific geographic region and target species, necessitating caution when extrapolating the findings to other areas or species [55]. These limitations underscore the need for further research to expand our understanding of species-environment interactions and to explore additional variables and modeling approaches that account for dynamic ecological processes.

## Conclusion

As keystone species, any change in the distribution of *Atta* species could lead to a mismatch of ecological function (*i.e.*, creating canopy gaps by transferring organic matter underground, *Atta* species contribute to enhancing soil aeration and increasing soil nutrients). Given their distribution, ants of the genus *Atta* are interesting from a biogeographic perspective, but their management presents a challenge. Understanding the factors and patterns that drive biodiversity across space and time is essential for effective conservation strategies for ecosystems and species. However, determining how habitats and species will respond to climate change is still an ongoing challenge, and filling these knowledge gaps is urgently needed.

## Supporting information

**S1 Dataset. Records of the species.** Occurrences of each species used to train the species distribution models.
(XLSX)

**S1 Table. Species ecoregions.** Ecoregions selected for each species based on known distribution: calibration using Olson et al.'s [21] terrestrial ecoregions model (M space).
(XLSX)

**S2 Table. Bioclimatic variables (names) used as predictors.** Bioclimatic variables (names) used as predictors in the species distribution models of three Mexican leaf-cutting ant species

(*A. cephalotes*, *A. mexicana*, and *A. texana*).
(DOCX)

## Acknowledgments

This study was developed under the project "Distribución espacial de *Atta mexicana* en respuesta al gradiente latitudinal urbano de la ciudad de Xalapa, Veracruz. México," registered at Universidad Veracruzana, No. 431112019124.

## Author Contributions

**Conceptualization:** Jorge A. Gómez-Díaz, Martha L. Baena, Christian A. Delfín-Alfonso.

**Data curation:** Jorge A. Gómez-Díaz, Martha L. Baena.

**Formal analysis:** Jorge A. Gómez-Díaz, Christian A. Delfín-Alfonso.

**Investigation:** Jorge A. Gómez-Díaz, Martha L. Baena, Arturo González-Zamora, Christian A. Delfín-Alfonso.

**Methodology:** Jorge A. Gómez-Díaz, Martha L. Baena, Arturo González-Zamora, Christian A. Delfín-Alfonso.

**Software:** Jorge A. Gómez-Díaz, Christian A. Delfín-Alfonso.

**Supervision:** Martha L. Baena.

**Validation:** Jorge A. Gómez-Díaz, Martha L. Baena, Christian A. Delfín-Alfonso.

**Visualization:** Jorge A. Gómez-Díaz, Christian A. Delfín-Alfonso.

**Writing – original draft:** Jorge A. Gómez-Díaz, Martha L. Baena, Arturo González-Zamora, Christian A. Delfín-Alfonso.

**Writing – review & editing:** Jorge A. Gómez-Díaz, Martha L. Baena, Arturo González-Zamora, Christian A. Delfín-Alfonso.

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
