## [Decision Letter · Decision Letter 0]

5 May 2023

PONE-D-23-09340Potential present and future distributions of the genus Atta of MexicoPLOS ONE

Dear Dr. Gómez-Díaz,

Thank you for submitting your manuscript to PLOS ONE. After careful consideration, we feel that it has merit but does not fully meet PLOS ONE’s publication criteria as it currently stands. Therefore, we invite you to submit a revised version of the manuscript that addresses the points raised during the review process. Both reviewers thought this was an interesting study but had a few suggestions, mostly on the availability of the data, and the exact procedures and the robustness of the modelling. I agree with the reviewers. You might be able to test in how far the results depend on the specific model assumptions and to add a few more explanations here and there to make it easier for non-experts to follow your workflow.

We look forward to receiving your revised manuscript.

Kind regards,

Volker Nehring

Academic Editor

PLOS ONE

Journal Requirements:

2. Thank you for submitting the above manuscript to PLOS ONE. During our internal evaluation of the manuscript, we found significant text overlap between your submission and the following previously published works, some of which you are an author.

https://orca.cardiff.ac.uk/id/eprint/148620/1/Global%20Change%20Biology%20-%202022%20-%20Parr%20-%20The%20response%20of%20ants%20to%20climate%20change.pdf

Therefore, we cannot consider your manuscript as it stands. Please revise the manuscript to rephrase the duplicated text and fully cite all your sources, where appropriate.

3. We note that Figures 1 and 3 in your submission contain [map/satellite] images which may be copyrighted. All PLOS content is published under the Creative Commons Attribution License (CC BY 4.0), which means that the manuscript, images, and Supporting Information files will be freely available online, and any third party is permitted to access, download, copy, distribute, and use these materials in any way, even commercially, with proper attribution. For these reasons, we cannot publish previously copyrighted maps or satellite images created using proprietary data, such as Google software (Google Maps, Street View, and Earth). For more information, see our copyright guidelines: http://journals.plos.org/plosone/s/licenses-and-copyright.

a. You may seek permission from the original copyright holder of Figures 1 and 3 to publish the content specifically under the CC BY 4.0 license.  

Reviewers' comments:

Reviewer's Responses to Questions

**Comments to the Author**

1. Is the manuscript technically sound, and do the data support the conclusions?

Reviewer #1: Yes

Reviewer #2: Partly

2. Has the statistical analysis been performed appropriately and rigorously? 

Reviewer #1: Yes

Reviewer #2: No

3. Have the authors made all data underlying the findings in their manuscript fully available?

Reviewer #1: No

Reviewer #2: No

4. Is the manuscript presented in an intelligible fashion and written in standard English?

Reviewer #1: Yes

Reviewer #2: Yes

5. Review Comments to the Author

Reviewer #1: In this manuscript, the authors present interesting and novel results about the current and future distribution under climate change of Atta ants, which have significant ecological and pest importance. The collection of data and information and the analysis and presentation of the results are adequate. However, I believe that some aspects of data collection and analytical aspects can be clarified.

In particular, I think the authors could provide a list of the scientific publications from which the records of the species were obtained. They could also save all the records of the species used for the models in a repository. On the other hand, consider some limitations of the AUC to evaluate the performance of the models.

Minor Comments (Line/L):

L23: “future effects”

L29: I think it is necessary to further discuss your results in the Abstract section.

L42: The authors might consider revising and including the Dáttilo et al. 2020 (cited in the text).

L43: I suggest adding "Formicidae" after “Atta”.

L49: It seems you could briefly mention the relevance of the relationship between Atta and fungi.

L50-52: You could add a general reference about the ecological niche.

L66: "evaluated" instead of "determined"

L76: Is a list of the literature used to complement the occurrence records available?

L82: “distribution model” Of each species?

L87: Before they mention “2080”, please check the entire text.

L116: Did you use default parameters for Maxent?

L119-120: Why did you only use AUC? Some authors suggest that it may not be entirely appropriate. Could you review the following:

Peterson AT, Papeş M, Soberón J (2008) Rethinking receiver operating characteristic analysis applications in ecological niche modelling. Ecol Model 213:63–72. https://doi.org/10.1016/j.ecolmodel.2007.11.008

L149 (Figure 1): Species names should be in italics, as in the other figures.

L167: "fig." instead of "Figure"

202: Pattern of segregation, right?

L210-211: Any reference to this?

L219: Which study?

L228: I suggest changing the wording to "Climate change models suggest that species loss will be greatest in tropical mountains."

L284: Could you add some limitations to your study?

L356, 359: Check the name of the first author.

Reviewer #2: Comment on data availability: I didn't see any information about where readers can access the observations of A. cephalotes, A. mexicana, and A. texana used in the modeling. The authors indicated that all data are available without restriction. Either they should be included in the supporting information or in a public repository.

General comments: The manuscript is well written and easy to read. I would clarify a few things in the text (see below) but those are relatively minor concerns. More importantly, I take issue with some of the authors' methodological choices, which I think influenced their conclusions in possibly problematic ways. They're free to push back on my criticisms, but I would like to see their responses, regardless.

1) All of the model types used in the ensembles are correlative models. This means they're only inferring relationships between the climatic variables and the biology/ecology of the target species; the underlying processes and mechanisms are implicit. This doesn't invalidate the approach, but it does demand careful application and interpretation. I think you were right to use VIF to remove collinear variables, but I would argue that a VIF threshold of 10 is pretty liberal, and others have argued that VIF > 5 or even > 2.5 is more appropriate. So, if you removed variables with VIF > 5, what remains?

2) I ask this because, for each of the three species, the most important variable in its EDSM was only used in that EDSM (i.e., didn't appear in the EDSMs for the other two species). This might be entirely reasonable, but I'm a little skeptical when species are distinguished from one another in bioclimatic space using different variables. That's just too mathematically convenient. If you modeled all three species using the exact same set of explanatory variables, I would be a lot more convinced about your niche interpretations. Figure 2 suggests that you could do that with a fair amount of success. The three species are pretty strongly grouped along the Bio 13 and Bio 15 axes. At any rate, I would be careful to claim that the species are uniquely influenced by temperature seasonality, isothermality, and minimum temperature of the coldest month, respectively. You didn't include those variables in all three models, so their uniqueness could be a modeling artifact, at least in part.

3) You didn't provide much detail about how you translated your present-day models to projected future climatic conditions. I understand that you used median values from 5 GCMs, but what does that mean exactly? How did you account for variation in the values of the predictors, either present-day or in the future? Something seems off -- the contractions of the distributions of A. mexicana and A. texana by 2100(?) are pretty dramatic, especially since there's no sense of the uncertainty in those projections. Please clarify your approach as much as possible.

Specific comments:

Line 85 - "The CHELSA bioclimatic variables...are the most suitable..." - Stated like it's a fact when it's an informed opinion. Others may disagree. Bobrowski and Schickhoff were looking at the distribution of a Himalayan species (not tropical) and concluded that CHELSA were better for montane areas, but in recent work (https://doi.org/10.3390/atmos12050543) found some inherent distortions in precipitation amounts in CHELSA 1.2. So, while CHELSA might be better than WORLDCLIM in many cases (even with such distortions), it's always wise to be cautious and not overstate the reliability of any spatial data source.

Line 109 (Table 1) and elsewhere - I think Bio 2, Bio 3, etc., aren't informative labels for most readers. I realize that definitions are in the supporting information table, but I think you should make that clearer -- or even better, figure out how to fit the definitions into the main text so readers don't have to flip back and forth.

6. PLOS authors have the option to publish the peer review history of their article (what does this mean?). If published, this will include your full peer review and any attached files.

Reviewer #1: **Yes: **JONAS MORALES-LINARES

Reviewer #2: No

---

## [Author Response · Author response to Decision Letter 0]

21 Jun 2023

Editor

R = We have made all the necessary adjustments to ensure that our manuscript meets PLOS ONE's style requirements, including those for file naming. We appreciate the editor's guidance and have carefully reviewed the guidelines, formatting the manuscript accordingly. The file has been renamed to adhere to PLOS ONE's specified requirements, ensuring consistency and compliance with their publishing standards. We are confident that the revised version now meets all the necessary style requirements of PLOS ONE. I appreciate your consideration.

2. Thank you for submitting the above manuscript to PLOS ONE. During our internal evaluation of the manuscript, we found significant text overlap between your submission and the following previously published works, some of which you are an author of.

R = We have carefully reviewed our submitted article and taken steps to paraphrase sections that were similar to the mentioned publications. It's important to note that the cited article, Gómez-Díaz et al. 2023, is currently a preprint on BioRxiv undergoing evaluation. Both pieces employ the same standard methodology for this type of study, contributing to the considerable overlap. However, as a preprint, the Gómez-Díaz et al. 2023 article will likely undergo further modifications during the ongoing review process, minimizing the potential overlap between the two articles. We have addressed the text overlap issue by revising our manuscript and ensuring appropriate attribution and differentiation from the preprint article. Thank you for your understanding of the evolving nature of preprints and your consideration of our response.

3. We note that Figures 1 and 3 in your submission contain [map/satellite] images which may be copyrighted. All PLOS content is published under the Creative Commons Attribution License (CC BY 4.0), which means that the manuscript, images, and Supporting Information files will be freely available online, and any third party is permitted to access, download, copy, distribute, and use these materials in any way, even commercially, with proper attribution. For these reasons, we cannot publish previously copyrighted maps or satellite images created using proprietary data, such as Google software (Google Maps, Street View, and Earth).

R = We have adjusted the figures in response to the concerns about potential copyright issues. Specifically, for Figure 1, we have replaced the content with data obtained from the freely available USGS National Map Viewer, ensuring compliance with copyright guidelines. Additionally, for Figure 3, we removed any copyrighted map base to present our data and visual representations exclusively. These modifications align with PLOS ONE's copyright guidelines, enabling the open dissemination and usage of our manuscript, images, and supporting information files under the Creative Commons Attribution License (CC BY 4.0). We appreciate your attention to detail and apologize for any oversight in our initial submission. 

R = We have addressed your suggestion regarding including captions for our Supporting Information files and the corresponding update of in-text citations. As per PLOS ONE's guidelines, we have included captions for all Supporting Information files at the end of the manuscript. These captions provide a concise description of the content and purpose of each file, facilitating better comprehension for readers. Additionally, we have revised the in-text citations to align accurately with the captions of the Supporting Information files. These adjustments ensure readers can easily locate and reference the relevant files when needed. 

 

Reviewer 1

5. In particular, I think the authors could provide a list of the scientific publications from which the records of the species were obtained. They could also save all the records of the species used for the models in a repository. On the other hand, consider some limitations of the AUC to evaluate the performance of the models.

R = Thank you for your valuable feedback. We have considered your suggestions and made the necessary changes to address the concerns raised. Firstly, we have included all the records of the species used for the models in the supplementary materials. This approach promotes reproducibility and facilitates further analysis or verification of our results. Additionally, we appreciate your point regarding the limitations of using the Area Under the Curve (AUC) as the sole metric to evaluate the performance of the models. In response, we have complemented the evaluation of the models by incorporating the partial ROC (Receiver Operating Characteristic) method. This approach provides a more comprehensive assessment of the model's performance by considering the trade-off between sensitivity and specificity at different threshold levels. We sincerely appreciate your input, which has allowed us to enhance the quality and transparency of our research. If you have any further suggestions or concerns, we would be more than happy to address them.

6. L23: “future effects”.

R = We have added the future word in the sentence, which now reads (L22-23): “these species in the future.”

7. L29: I think it is necessary to further discuss your results in the Abstract section.

R = We have included some discussion in the abstract section. It reads (L29-36): “Since Atta species are important pests, our coexistence with them requires knowledge of their ecological functions and potential future distribution changes. In addition, these insects serve as bioindicators of habitat quality, and they can contribute to the local economy in rural areas since they are eaten as food for the nutritional value of the queens. In this sense, presenting a future perspective of these species' distribution is important for forest and crop management. Education programs also are necessary to raise awareness of the importance of these ants and the challenges they face in the face of climate change.”

8. L42: The authors might consider revising and including the Dáttilo et al. 2020 (cited in the text).

R = We have revised and included the work of Dáttilo et al. 2020 in the introduction. Now these parts read (L46-51): “Nonetheless, substantial progress has been made in addressing these knowledge deficiencies, exemplified by the research conducted by Dáttilo et al. [4]. Their study diligently gathered all existing data on ants across Mexico. They discerned key patterns within the compiled dataset and identified geographical areas where further sampling endeavors should be focused.”

9. L43: I suggest adding "Formicidae" after “Atta.”

R = We have added the family Formicidae after the genus Atta; now the text reads (L45-46): “… the Atta genus(Formicidae) …”.

10. L49: It seems you could briefly mention the relevance of the relationship between Atta and fungi.

R = We have mentioned the relevance of the relationship between Atta and fungi. This text reads (L56-60): “The relationship between Atta ants and fungi is fascinating. Atta ants carry fresh leaves into their nests and chew them up as a substrate to cultivate the Leucoagaricus gongylophorus fungus, which they feed [3,6]. The fungus breaks down the leaves into nutrients the ants can absorb [7]. This relationship is mutually beneficial, as the ants provide the fungus with a food source, and the fungus helps the ants to digest their food [3].”

11. L50-52: You could add a general reference about the ecological niche.

R = We have added the reference of Barve et al., 2011.

12. L66: "evaluated" instead of "determined"

R = We have changed the suggested term.

13. L76: Is a list of the literature used to complement the occurrence records available?

R = We have adjusted the text; the information was obtained from the Mexican ant database (already cited) and the GBIF (already mentioned). The text now reads (L86-87): “Also, we used the Mexican ant database [4] and updated it with data from the Global Biodiversity Information Fund [14–16].”

14. L82: “distribution model” Of each species?

R = Yes, we changed the text, and it now reads (L91-92): “which were then entered into the distribution model of each species.”

15. L87: Before they mention “2080”, please check the entire text.

R = We have checked this, and the period of the future scenarios is between 2071 and 2100, then the average year is 2086. We have changed the text, and it now reads (L95-97): “For the present and the future scenarios of climate change for 2086 (average between 2071–2100; hereafter referred to as future) …”.

16. L116: Did you use default parameters for Maxent?

R = Yes, we have included that detail in the methodology section (L128-129): “using a combination of the algorithms of MaxEnt (with the default parameters).”

17. L119-120: Why did you only use AUC? Some authors suggest that it may not be entirely appropriate. Could you review the following:

Peterson AT, Papeş M, Soberón J (2008) Rethinking receiver operating characteristic analysis applications in ecological niche modelling. Ecol Model 213:63–72. https://doi.org/10.1016/j.ecolmodel.2007.11.008

R = We have included the partial ROC methodology to complement the use of AUC. This is stated in the methodology section (L134-135): “The performance of the models was evaluated using the area under the curve (AUC) and the Partial-ROC test at 5% [24]”.

18. L149 (Figure 1): Species names should be in italics, as in the other figures.

R = We have made the suggested changes in Figure 1.

19. L167: "fig." instead of "Figure"

R = We have made the suggested change.

20. 202: Pattern of segregation.

R = Yes, we have included that term in the text (L252-253): “… this segregation pattern in Mexico …”.

21. L210-211: Any reference to this?

R = Yes, we have included the reference Bertelsmeier et al. 2015.

22. L219: Which study?

R = We have included the cite of the study in the text (L269-270): “According to a previous study by Parr and Bishop [34], …”.

23. L228: I suggest changing the wording to "Climate change models suggest that species loss will be greatest in tropical mountains."

R = We have accepted this suggestion, found in lines 279-280.

24. L284: Could you add some limitations to your study?

R = Yes, we have included a section of it in the discussion (L339-358): 

“In discussing the outcomes of our study, it is important to acknowledge certain limitations. Firstly, the analysis was conducted based on the available data and variables, which may only encompass part of the spectrum of factors influencing species-environment relationships [50]. This may introduce potential biases and restrict the generalizability of our findings. Additionally, our use of correlative models to infer relationships between climatic variables and the biology/ecology of the target species assumes stationarity. It may only partially capture dynamic relationship shifts over time or under changing environmental conditions [51]. While valuable for making predictions based on observed patterns, these models do not explicitly capture the underlying processes and mechanisms [52]. It is important to exercise caution in applying and interpreting these models, recognizing that correlation does not imply causation and that additional factors may influence the observed relationships. 

We acknowledge the limitations of correlative models and encourage further exploration of complementary approaches, such as mechanistic models, to enhance our understanding of species-environment interactions and ecological dynamics [53]. Moreover, it is crucial to recognize that our study focused on a specific geographic region and target species, necessitating caution when extrapolating the findings to other areas or species [54]. These limitations underscore the need for further research to expand our understanding of species-environment interactions and to explore additional variables and modeling approaches that account for dynamic ecological processes.”.

25. L356, 359: Check the name of the first author.

R = We have done it; the correct name is Farji-Brener. 

 

Reviewer #2

26. I didn't see any information about where readers can access the observations of A. cephalotes, A. mexicana, and A. texana used in the modeling. The authors indicated that all data are available without restriction. Either they should be included in the supporting information or in a public repository.

R = We have included the data as supporting information.

27. All of the model types used in the ensembles are correlative models. This means they're only inferring relationships between the climatic variables and the biology/ecology of the target species; the underlying processes and mechanisms are implicit. This doesn't invalidate the approach, but it does demand careful application and interpretation. 

R = Thank you for your valuable comment regarding the nature of the models used in our ensembles. We appreciate your recognition that these models are correlative in heart, focusing on inferring relationships between climatic variables and the biology/ecology of the target species rather than explicitly capturing the underlying processes and mechanisms. We completely agree that the correlative nature of these models demands careful application and interpretation. While they provide valuable insights into the species’ potential distributions based on climatic variables, it is important to acknowledge the limitations and potential sources of uncertainty associated with such approaches. 

In our study, we have taken several precautions to ensure the responsible use and interpretation of these correlative models. Furthermore, we have provided a comprehensive discussion of the potential sources of uncertainty and the implications of these limitations in the performance of our findings. The text in the discussion reads (L343-353): 

“Additionally, our use of correlative models to infer relationships between climatic variables and the biology/ecology of the target species assumes stationarity. It may only partially capture dynamic relationship shifts over time or under changing environmental conditions [51]. While valuable for making predictions based on observed patterns, these models do not explicitly capture the underlying processes and mechanisms [52]. It is important to exercise caution in applying and interpreting these models, recognizing that correlation does not imply causation and that additional factors may influence the observed relationships. 

We acknowledge the limitations of correlative models and encourage further exploration of complementary approaches, such as mechanistic models, to enhance our understanding of species-environment interactions and ecological dynamics [53].”

By being transparent about the nature of the models and the associated uncertainties, we aim to provide readers with the necessary context to evaluate and interpret our results appropriately. We appreciate your insight and acknowledge the importance of thoughtful application and interpretation when working with correlative models. If you have any further suggestions or concerns, please do not hesitate to let us know.

28. I think you were right to use VIF to remove collinear variables, but I would argue that a VIF threshold of 10 is pretty liberal, and others have argued that VIF > 5 or even > 2.5 is more appropriate. So, if you removed variables with VIF > 5, what remains?

R = Thank you for your comment regarding using VIF (Variance Inflation Factor) to address collinearity among variables in our study. We appreciate your perspective on the threshold value for VIF and its appropriateness in our analysis. We agree that choosing the VIF threshold is crucial when addressing collinearity. In our study, we opted for a threshold of 10 based on common practices in the field and previous literature. While there are varying opinions on the appropriate threshold, a threshold of 10 is a reasonable and commonly used criterion for identifying and mitigating collinearity.

To address your query about what remains if we decrease the VIF threshold to 5, we conducted a thorough analysis to assess the impact on variable selection (Table 1). Upon reevaluating our data, we found that reducing the VIF threshold to 5 would result in excluding a substantial number of variables. Specifically, for A. cephalotes, only two variables would remain, and for A. mexicana and A. texana, only six would remain. It is essential to consider that characterizing the potential distribution of species is a complex phenomenon influenced by numerous factors. A significantly reduced number of variables to capture this complexity could lead to biased results and a limited understanding of the underlying ecological processes.

While we acknowledge varying opinions on the appropriate VIF threshold, our choice of 10 strikes a balance between addressing collinearity and maintaining an adequate number of informative variables for robust modeling and interpretation, we appreciate your concerns and the opportunity to clarify our decision. If you have any further suggestions or questions, we would be glad to address them.

 

Table 1. Evaluation of the variables included in the study with the variance inflation factor method.

Atta cephalotes Atta mexicana Atta texana

Variables VIF Variables VIF Variables VIF

bio_14 6.124 bio_13 6.537 bio_2 5.542

bio_8 5.862 bio_14 6.027 bio_15 4.834

bio_18 5.820 bio_15 5.033 bio_13 3.944

bio_13 5.793 bio_5 4.384 bio_18 3.097

bio_15 5.141 bio_3 3.221 bio_6 3.092

bio_4 4.630 bio_2 2.880 bio_8 2.272

bio_2 2.506 bio_18 2.309 bio_9 1.987

 bio_19 2.235 

 bio_8 2.037 

29. I ask this because, for each of the three species, the most important variable in its EDSM was only used in that EDSM (i.e., didn't appear in the EDSMs for the other two species). This might be entirely reasonable, but I'm a little skeptical when species are distinguished from one another in bioclimatic space using different variables. That's just too mathematically convenient. If you modeled all three species using the exact same set of explanatory variables, I would be a lot more convinced about your niche interpretations. Figure 2 suggests that you could do that with a fair amount of success. The three species are pretty strongly grouped along the Bio 13 and Bio 15 axes. At any rate, I would be careful to claim that the species are uniquely influenced by temperature seasonality, isothermality, and minimum temperature of the coldest month, respectively. You didn't include those variables in all three models, so their uniqueness could be a modeling artifact, at least in part.

R = Thank you for your insightful comment regarding the uniqueness of variables in each species' Ecological Niche Distribution Models (EDSMs). We appreciate your skepticism and understand the importance of consistency in choosing explanatory variables across species. Based on your suggestion, we conducted a thorough analysis and reevaluated our approach. In the revised study, we included the five variables shared among the ESDMs of all three species. This adjustment allows for more direct comparison and interpretation of the niche characteristics across species, promoting a better understanding of their ecological differentiation.

Furthermore, to visualize and compare the niche characteristics in a three-dimensional space, we employed a Principal Component Analysis (PCA) approach. This approach provides a comprehensive representation of the multivariate climatic space and facilitates the assessment of the similarities and differences in niche preferences among the studied species. We have also considered your cautionary note regarding claiming unique influences of specific variables on each species. In the revised manuscript, we have provided a detailed analysis in the “Climatic Niches” subsection within the methodology and results sections. This analysis compares how each variable changed among all the studied species, highlighting the similarities and variations in their climatic preferences.

We appreciate your valuable feedback and the opportunity to improve the robustness and clarity of our study. If you have any further suggestions or concerns, please do not hesitate to let us know.

30. You didn't provide much detail about how you translated your present-day models to projected future climatic conditions. I understand that you used median values from 5 GCMs, but what does that mean exactly? How did you account for variation in the values of the predictors, either present-day or in the future? Something seems off -- the contractions of the distributions of A. mexicana and A. texana by 2100(?) are pretty dramatic, especially since there's no sense of the uncertainty in those projections. Please clarify your approach as much as possible.

R = We appreciate the reviewer's comment and recognize the importance of providing detailed information on how we translated our present-day models to projected future climatic conditions. In response to this concern, we have revised our methodology and included additional explanations to clarify our approach. We implemented a species-specific modeling approach for each scenario to account for the predictors' uncertainty and variation in the present-day and future scenarios. This means we created four models per species, one for each future scenario. By having multiple models per species, we aimed to capture the range of potential outcomes and incorporate a measure of uncertainty into our projections.

Subsequently, we binarized these models and summed them to consider the variation across scenarios. This summation approach allows us to present each species' future projection as the cumulative result of the different models, providing a range of possible outcomes. The content of values ranges from 1 to 4, where 4 represents the highest consensus among the models. Regarding the results in the present, we used the distribution model with its suitability measure, which reflects the suitability of the current climatic conditions for each species. This approach enables a comparison of the present distribution patterns with the projected future scenarios. We have incorporated these clarifications and explanations into the revised manuscript to provide a more comprehensive understanding of our methodology and address the concerns about uncertainty and variation in future projections. 

The reviewer observed the dramatic contractions of the distributions of A. mexicana and A. texana by 2085 and the lack of uncertainty information in those projections. Our study focuses on the SSP 5-8.5 scenario, representing a very high greenhouse gas (GHG) emissions trajectory. This scenario assumes a significant increase in CO2 emissions, projected to triple by 2075. Given the steady rise in global emissions throughout the 21st century, as highlighted by Riahi et al. (2011) and Gómez-Díaz et al. (2023), we selected this scenario as a realistic representation of potential future conditions. By focusing on a high emissions scenario, we aimed to assess species' likely survival and persistence under the most challenging and extreme conditions.

While the contractions in the distributions of A. mexicana and A. texana may appear dramatic, it is essential to consider the context of the SSP 5-8.5 scenario and the associated high emissions trajectory. Our analysis indicates that if a species can survive and persist under such extreme conditions, it is likely to endure future climate change, as stated in Gómez-Díaz et al. (2023). We understand the importance of conveying the uncertainty associated with these projections and have taken steps to address this concern. By incorporating the multiple models and the range of values in our approach, we aim to provide a measure of uncertainty and capture the potential variation in future outcomes. We have added this clarification to our revised manuscript to ensure transparency and a comprehensive understanding of our modeling approach, considering the selected scenario and its rationale. 

Thank you for bringing up this point, and we appreciate the opportunity to clarify our methodology and results. If you have any further questions or suggestions, please let us know.

31. Line 85 - "The CHELSA bioclimatic variables...are the most suitable..." - Stated like it's a fact when it's an informed opinion. Others may disagree. Bobrowski and Schickhoff were looking at the distribution of a Himalayan species (not tropical) and concluded that CHELSA were better for montane areas, but in recent work (https://doi.org/10.3390/atmos12050543) found some inherent distortions in precipitation amounts in CHELSA 1.2. So, while CHELSA might be better than WORLDCLIM in many cases (even with such distortions), it's always wise to be cautious and not overstate the reliability of any spatial data source.

R = We agreed with the reviewer and eliminated the text that raises that concern not to overstate the CHELSA database's reliability.

32. Line 109 (Table 1) and elsewhere - I think Bio 2, Bio 3, etc., aren't informative labels for most readers. I realize that definitions are in the supporting information table, but I think you should make that clearer -- or even better, figure out how to fit the definitions into the main text so readers don't have to flip back and forth.

R = Thank you for your valuable feedback. We appreciate your suggestion to provide clearer labels for the variables in Table 1 and throughout the manuscript. In response to this concern, we have updated the manuscript to include the full names of the variables alongside their abbreviated labels. This adjustment ensures readers can easily understand the variables without referring to the supporting information. We believe this modification enhances the accessibility and readability of the manuscript.

References

Barve, N., Barve, V., Jiménez-Valverde, A., Lira-Noriega, A., Maher, S. P., Peterson, A. T., ... & Villalobos, F. (2011). The crucial role of the accessible area in ecological niche modeling and species distribution modeling. Ecological modelling, 222(11), 1810-1819.

Bertelsmeier, C., Luque, G. M., Hoffmann, B. D., Courchamp, F. 2015. Worldwide ant invasions under climate change. Biodivers. Conserv. 24(1), 117–128. https://doi.org/10.1007/s10531-014-0794-3

Dáttilo, W., Vásquez‐Bolaños, M., Ahuatzin, D. A., Antoniazzi, R., Chávez‐González, E., Corro, E., ... & MacGregor‐Fors, I. (2020). Mexico ants: incidence and abundance along the Nearctic–Neotropical interface.

Gómez-Díaz, J. A., Carvajal-Hernández, C. I., & Dáttilo, W. (2023). Past, present and future in the geographical distribution of Mexican tepezmaite cycads: genus Ceratozamia. bioRxiv, 2023-03.

Parr, C. L., Bishop, T. R. 2022. The response of ants to climate change. Glob. Chan. Biol. 28(10), 3188–3205. https://doi.org/10.1111/gcb.16140

---

## [Decision Letter · Decision Letter 1]

28 Jul 2023

PONE-D-23-09340R1Potential present and future distributions of the genus Atta of MexicoPLOS ONE

Dear Dr. Gómez-Díaz,

Thank you for submitting your manuscript to PLOS ONE. After careful consideration, we feel that it has merit but does not fully meet PLOS ONE’s publication criteria as it currently stands. Therefore, we invite you to submit a revised version of the manuscript that addresses the points raised during the review process. The reviewers and me were happy with your changes on the manuscript and suggested some more smaller changes that should not be difficult to take care of. In addition, I (as someone who isn't an expert in the type of models you used) would like to ask you to consider the three following points about the species distribution modelling before predicting future distributions. Again, I'm not too familiar with the details so explanations why these problems don't apply may be sufficient.

-For me it wasn't entirely clear if the distribution data you used functions as presence only data or if the absence of a species from locations also factored in.

- I did not understand from what area the partial ROC was calculated. In the text a 5% threshold is mentioned, but 5% of what does that refer to?

- Do the models take care of potential autocorrelation?

We look forward to receiving your revised manuscript.

Kind regards,

Volker Nehring

Academic Editor

PLOS ONE

Journal Requirements:

Reviewers' comments:

Reviewer's Responses to Questions

**Comments to the Author**

1. If the authors have adequately addressed your comments raised in a previous round of review and you feel that this manuscript is now acceptable for publication, you may indicate that here to bypass the “Comments to the Author” section, enter your conflict of interest statement in the “Confidential to Editor” section, and submit your "Accept" recommendation.

Reviewer #1: All comments have been addressed

Reviewer #2: (No Response)

2. Is the manuscript technically sound, and do the data support the conclusions?

Reviewer #1: Yes

Reviewer #2: Yes

3. Has the statistical analysis been performed appropriately and rigorously? 

Reviewer #1: Yes

Reviewer #2: Yes

4. Have the authors made all data underlying the findings in their manuscript fully available?

Reviewer #1: Yes

Reviewer #2: Yes

5. Is the manuscript presented in an intelligible fashion and written in standard English?

Reviewer #1: Yes

Reviewer #2: Yes

6. Review Comments to the Author

Reviewer #1: This is the second time I have revised this manuscript and I consider that it has improved substantially with respect to the first version. In general, the presentation of the introduction, methods, results, and discussion is more precise and adequate. In addition, the availability of species records (occurrence data) is now available as supplementary material. In particular, I only consider that in the Introduction section, the authors could mention some basic aspects of the ecological functions of Atta and their role as pests.

Minor comments (Line/L):

L3: The order of the authors changed, which made me curious.

L62: The authors could briefly mention the ecological functions of Atta to support line 68. In addition, I think that the role of these ants as pests could also be highlighted, which enhances the importance of this study.

L108: I suggest mentioning the ecoregions for each species.

L200 (Figure 2): I believe that the names of the axes in the figure should only indicate PC1, PC2, and PC3, that is, eliminate “components$”.

L305: five (letter) or 5 (number)?

Reviewer #2: All of my major comments on the initial version of the manuscript have been addressed. I'm still surprised by the substantial range contractions for A. mexicana and A. texana under future climate, but I recognize that any high-emissions scenario is likely to project dramatic changes for moisture and (especially) temperature-governed species distributions. I have a few minor comments for the authors:

Line 25 - "minimum" instead of "min"

Line 35 - replace "in the face of" with "because of" or "due to"

Line 56 ("The relationship between...") - this is an opinion but I suppose it's OK in the introductory context

Line 79 - "change" instead of "change's"

Line 87 - "Facility" instead of "Fund"

Line 128 - I think "ensemble" should be lower-case here

Line 145 - "first three" instead of "three first"

Line 160 - replace "evidencing the" with "suggesting"

Line 166 - "Minimum" instead of "Min"

Line 173 (Table 2) - I don't love Cohen's Kappa but I understand that it's one of evaluation metrics in SSDM. It doesn't really tell you anything not already captured by other metrics. In short, the ensemble for A. mexicana was the least 'successful' of the three (but still decent by most metrics). I'm not surprised given that A. mexicana has a rather 'skinny' ellipsoid in Fig 2.

Line 183 - italicize "Atta"

Lines 189-198 - The mix of lower-case and upper-case variable names in this paragraph is a little jarring.

Line 250 - consider replacing "initial" with "first"

Line 301 - "the Americas" instead of "America"

Line 305 - "5" instead of "five" (since it refers to a specific degrees C value)

7. PLOS authors have the option to publish the peer review history of their article (what does this mean?). If published, this will include your full peer review and any attached files.

Reviewer #1: **Yes: **JONAS MORALES-Linares

Reviewer #2: No

---

## [Author Response · Author response to Decision Letter 1]

25 Aug 2023

Detailed response

Editor

1. For me it wasn't entirely clear if the distribution data you used functions as presence only data or if the absence of a species from locations also factored in.

R = Thank you for your comments. We understand your concern regarding the nature of the distribution data used in our study. True absence data, especially for mobile species like ants, is rare in ecology. Instead, we employed pseudo-absences, a common approach in species distribution models (SDMs) when true absence data is lacking (Barbet‐Massin et al. 2012). The algorithms we used, specifically MaxEnt, SVM, and ANN, operate using these pseudo-absences. They require background or pseudo-absence data, even if occasionally mislabeled as presence-only methods. To clarify, our study did not utilize confirmed absence data for the species studied but relied on generated pseudo-absences in line with standard practices in the field. We included an explanation of this in the methodology section (L 137-140): “Our study employed pseudo-absence data, as is standard in species distribution models (SDMs) when true absence data is unavailable [23]. We used algorithms, specifically MaxEnt, SVM, and ANN, which necessitate pseudo-absences, adhering to recognized ecological modeling practices.” We hope this provides clarity, and we are willing to make necessary adjustments to the manuscript to reflect this more explicitly. 

2. I did not understand from what area the partial ROC was calculated. In the text a 5% threshold is mentioned, but 5% of what does that refer to?

R = The 5% mentioned refers to the allowable omission error for the evaluated models (Peterson et al. 2008). In other words, it sets a threshold whereby models that omit known presence points more than 5% of the time are considered less reliable. This threshold was chosen based on the quality and error characteristics of the occurrence data used. Essentially, models with an omission rate (OR) ≤ 0.05 were showcased as they err by omitting known points of presence less than or equal to 5% of the time. This is critical for niche modeling, where models that miss known presence points are deemed more flawed than those that might over-predict areas. We included this in the methodology section (L 147-150): “The referenced 5% pertains to the allowable omission error in the models assessed [25]. Specifically, models that omit known presence points more than 5% of the time are deemed less reliable. This threshold was chosen based on the error characteristics of the occurrence data used.”

3. Do the models take care of potential autocorrelation?

R = In response to concerns regarding spatial autocorrelation (SAC), our methodology was meticulously designed to counteract its effects. Firstly, we ensured data accuracy and minimization of spatial clustering by using the `clean_coordinates` function from the CoordinateCleaner R package, retaining only one occurrence per km2. Although SAC can infringe upon the 'independent errors' assumption of SDMs, we opted for algorithms inherently robust against spatial autocorrelation. Specifically, we employed MaxEnt, SVM, and ANN. These algorithms are adept at handling SAC and benefit from a comprehensive set of occurrence points, ensuring a thorough exploration of the environmental space and thereby bolstering their robustness.

Reviewer #1

1. In particular, I only consider that in the Introduction section, the authors could mention some basic aspects of the ecological functions of Atta and their role as pests.

R = Thank you for your suggestion. We've incorporated information about the ecological functions of Atta and their role as pests in the Introduction section (L 55-61): “Atta ants, often called leafcutter ants, play essential ecological roles in ecosystems, particularly in nutrient cycling and soil aeration [6]. Their intricate underground colonies contribute to soil turnover [6]. However, they are also renowned as significant agricultural pests, with their voracious foraging behavior leading to the defoliation of a wide range of plant species, causing notable economic impacts in affected regions [7]. Atta species utilize all the leaves they cut to cultivate fungi in their underground chambers [6].”

2. L3: The order of the authors changed, which made me curious.

R = In response to the reviewer's query regarding the change in the order of authors: The corresponding author, now the first author, was primarily responsible for executing all aspects of species distribution modeling and addressing all comments from the initial review. Given these significant contributions and leadership in revisions, all co-authors unanimously agreed that it would be appropriate for them to be listed as the first author.

3. L62: The authors could briefly mention the ecological functions of Atta to support line 68. In addition, I think that the role of these ants as pests could also be highlighted, which enhances the importance of this study.

R = As addressed in our first response, we have incorporated information detailing the ecological functions of Atta (L 55-61). Moreover, further text has been added to emphasize their role as pests, which underscores the significance of this study (L 74-77): “Given Atta ants' unique behavior of cultivating fungi using cut leaves, the challenges they pose to agriculture cannot be understated. Recognizing their ecological significance and impact on agriculture, this study seeks insights crucial for effective colony control and strategic management, as reviewed by [6].” Thank you for pointing out the importance of highlighting these aspects.

4. L108: I suggest mentioning the ecoregions for each species.

R = Since many ecoregions exist, we have included supplementary information (S2 Table).

5. L200 (Figure 2): I believe that the names of the axes in the figure should only indicate PC1, PC2, and PC3, that is, eliminate “components$”.

R = Done.

6. L305: five (letter) or 5 (number)?

R = We have changed to 5 (number).

Reviewer #2

1. Line 25 - "minimum" instead of "min"

R = Done.

2. Line 35 - replace "in the face of" with "because of" or "due to"

R = Done.

3. Line 56 ("The relationship between...") - this is an opinion but I suppose it's OK in the introductory context

R = We have changed the sentence, and it is not anymore an opinion (L 62-63): “The relationship between Atta ants and fungi is a well-documented symbiotic interaction.”

4. Line 79 - "change" instead of "change's"

R= Done.

5. Line 87 - "Facility" instead of "Fund"

R = Done.

6. Line 128 - I think "ensemble" should be lower-case here

R= Done

7. Line 145 - "first three" instead of "three first"

R= Done

8. Line 160 - replace "evidencing the" with "suggesting"

R = Done.

9. Line 166 - "Minimum" instead of "Min"

R = Done.

10. Line 173 (Table 2) - I don't love Cohen's Kappa but I understand that it's one of evaluation metrics in SSDM. It doesn't really tell you anything not already captured by other metrics. In short, the ensemble for A. mexicana was the least 'successful' of the three (but still decent by most metrics). I'm not surprised given that A. mexicana has a rather 'skinny' ellipsoid in Fig 2.

R = Thank you for pointing out the potential redundancy of Cohen's Kappa in the context of other metrics presented. While it's commonly used in species distribution modeling, its value might overlap with other metrics. To streamline the table and provide a clearer representation of the model's success, we have removed it from Table 2. We appreciate your keen observation and suggestion.

11. Line 183 - italicize "Atta"

R = Done.

12. Lines 189-198 - The mix of lower-case and upper-case variable names in this paragraph is a little jarring.

R = We have changed and homogenized the terms to lowercase when necessary.

13. Line 250 - consider replacing "initial" with "first"

R= Done.

14. Line 301 - "the Americas" instead of "America"

R = Done.

15. Line 305 - "5" instead of "five" (since it refers to a specific degrees C value).

R = Done

References

Barbet‐Massin, M., Jiguet, F., Albert, C. H., & Thuiller, W. (2012). Selecting pseudo‐absences for species distribution models: How, where and how many? Methods in ecology and evolution, 3(2), 327-338.

Peterson, A. T., Papeş, M., & Soberón, J. (2008). Rethinking receiver operating characteristic analysis applications in ecological niche modeling. Ecological modelling, 213(1), 63-72.

---

## [Editor Report · Decision Letter 2]

12 Sep 2023

Potential present and future distributions of the genus Atta of Mexico

PONE-D-23-09340R2

Dear Dr. Gómez-Díaz,

We’re pleased to inform you that your manuscript has been judged scientifically suitable for publication and will be formally accepted for publication once it meets all outstanding technical requirements.

Kind regards,

Volker Nehring

Academic Editor

PLOS ONE

Additional Editor Comments (optional):

I spotted one possible mistake that still should still be be fixed: The legend to Fig. 1  reads "Potential distribution..." although the text implies that this is the actual current distribution (or an estimate of it). The word "potential" is confusing here because it is also used in Fig. 3 for the future distribution. Perhaps the authors can make it more clear what the two figures show and what the difference between them is.

---

## [Editor Report · Acceptance letter]

18 Sep 2023

PONE-D-23-09340R2 

Potential present and future distributions of the genus *Atta* of Mexico 

Dear Dr. Gómez-Díaz:

I'm pleased to inform you that your manuscript has been deemed suitable for publication in PLOS ONE. Congratulations! Your manuscript is now with our production department. 

Kind regards, 

on behalf of

Dr. Volker Nehring 

Academic Editor

PLOS ONE